# Predicting Colorectal Cancer Using Machine and Deep Learning Algorithms: Challenges and Opportunities

Dabiah Alboaneen [1,*], Razan Alqarni [1], Sheikah Alqahtani [1], Maha Alrashidi [1], Rawan Alhuda [1], Eyman Alyahyan [1] and Turki Alshammari [2,3]

1 Computer Science Department, College of Sciences and Humanities, Imam Abdulrahman Bin Faisal University, Jubail 31961, Saudi Arabia
2 Colorectal Surgery Unit, Department of Surgery, King Fahad Specialist Hospital-Dammam, Dammam 31444, Saudi Arabia
3 College of Medicine, Imam Abdulrahman Bin Faisal University, Dammam 31441, Saudi Arabia
* Correspondence: dabuainain@iau.edu.sa

**Abstract:** One of the three most serious and deadly cancers in the world is colorectal cancer. The most crucial stage, like with any cancer, is early diagnosis. In the medical industry, artificial intelligence (AI) has recently made tremendous strides and showing promise for clinical applications. Machine learning (ML) and deep learning (DL) applications have recently gained popularity in the analysis of medical texts and images due to the benefits and achievements they have made in the early diagnosis of cancerous tissues and organs. In this paper, we intend to systematically review the state-of-the-art research on AI-based ML and DL techniques applied to the modeling of colorectal cancer. All research papers in the field of colorectal cancer are collected based on ML and DL techniques, and they are then classified into three categories: the aim of the prediction, the method of the prediction, and data samples. Following that, a thorough summary and a list of the studies gathered under each topic are provided. We conclude our study with a critical discussion of the challenges and opportunities in colorectal cancer prediction using ML and DL techniques by concentrating on the technical and medical points of view. Finally, we believe that our study will be helpful to scientists who are considering employing ML and DL methods to diagnose colorectal cancer.

**Keywords:** artificial intelligence; colorectal cancer; deep learning; early diagnosis; machine learning

## 1. Introduction

Nowadays, the number of patients with cancer disease has been increasing in the world, depending on many factors such as eating unhealthy foods, obesity, genetic inheritance, and age [1]. On the other hand, in our bodies, cells have instructions about how to divide and grow. Deoxyribonucleic acid (DNA) has these instructions in the form of genes. Cells do not stop growing when there are mutations that cause malignant tumors (cancer) that grow uncontrollably and irregularly. Colorectal cancer is deadly and can be considered the third major cause of cancer-related deaths around the world [2,3] and is the third most common type of cancer in Saudi Arabia [4]. A total of 90% of colorectal cancer cases occur in people aged above 45 [5], and men are at the top of the term rank of incidence in colorectal cancer [6]. Colorectal cancer is one of the most serious malignant tumors. It is the cause of death for over 4000 people annually in the Kingdom of Saudi Arabia [7].

According to the Saudi Ministry of Health (MoH), early and periodic screening for colorectal cancer is based on the patient's history and symptoms. The target group in the early detection of colorectal cancer is people with low risk, who are between the ages of 45–75, and people with high risk, who have a previous history of cancer or a family history of the disease, or who were exposed to radiation therapy during childhood. The col-

orectal examination follows two types, which are the fecal occult blood test (FOBT)/fecal immunological test (FIT), and total colonoscopy [6].

Artificial intelligence (AI) has enhanced human life in the healthcare sector which has led to an increase in the quality and efficiency of the performance of systems and services in health areas [8]. AI can be used as a solution to predict colorectal cancer by using machine learning (ML) and deep learning (DL) algorithms. A prediction model using ML for predicting colorectal cancer helps in obtaining a faster, more accurate diagnosis of colorectal cancer in its early stage, increases the success rate of treatment, and reduces the colorectal cancer mortality rate [9]. There is a diversity of goals for predicting colorectal cancer disease. It is possible to predict the presence of cancer in its initial stages or identify the stage of cancer to determine the appropriate type of treatment and treatment plan, in addition to enhancing the lifestyle quality of colorectal cancer patients [10].

Several research studies have conducted a review of previous research on AI-based colorectal cancer diagnosis. On the topic of colorectal cancer and DL, Pacal et al. [3] provided an overview of colorectal cancer-related DL structures. In addition to analyzing DL publications, they classified them into five categories: detection, classification, segmentation, survival prediction, and inflammatory bowel diseases.

Furthermore, Ref. [11] presented an in-depth view of recently published research publications on colorectal cancer diagnosis and prognosis based on DL techniques. The authors emphasize some outstanding issues and provide some insights into the feasibility and development of robust diagnostic systems for future health care and oncology.

Using digital image analysis on histopathological images, Ref. [12] provides a systematic review of the application of DL in colorectal cancer. Additionally, the limitations were outlined in order to encourage researchers to provide solutions.

In terms of colorectal cancer and ML, Kourou et al. [13] reviewed studies that applied ML algorithms to cancer prediction. ML has proven to be extremely efficient in cancer prediction.

Although the medical domain has shown a lot of interest in the use of DL and ML to diagnose cancer, comprehensive literature reviews that cover all aspects of colorectal cancer diagnosis and prognosis utilizing cutting-edge DL and ML methods are still limited.

Therefore, this paper presents a comprehensive review of the previous contributions achieved by researchers in the prediction of colorectal cancer based on both ML and DL algorithms. The search for these papers is conducted in three stages. The majority of papers have been published since 2011, and they are tabulated and sorted. They are investigated from several perspectives, including the aim, the method, and the dataset. The goal of the paper is to investigate colorectal cancer from both medical and technical viewpoints. Additionally, it emphasizes a number of challenges and opportunities in predicting colorectal cancer.

The remainder of this paper is arranged as follows. In Section 2, a brief background on cancer disease, colorectal cancer, and AI is presented. In Section 3, the filtered research articles that have been determined to be selected research are included. Section 4 presents a literature review and, accordingly, contains an analysis of models and theoretical frameworks that have been previously introduced to the research area. Different performance metrics are mentioned in Section 5. The most widely used aims, algorithms, and data types are in Section 6. In addition, there are research gaps. Finally, Section 7 concludes the work.

## 2. Background

### 2.1. Cancer Disease

One of the most critical illnesses is cancer. Human bodies are formed of cells. There are instructions on how to divide and grow every cell. DNA contains these instructions in the form of genes. DNA is passed on to new cells when a cell divides. It is possible to make mistakes when copying DNA, known as mutations. Based on the instructions, DNA usually tells cells when to stop growing and dividing. However, in the case of cells

having mutations in their DNA that give instructions on when to stop growing, a cell may continue to grow. This is how cancer starts to form.

Cancer can occur in any part of the body, such as the brain, lungs, breast, colon, rectum, liver, and even the blood. The tumor happens when the cells divide out of control, which makes a clump of cells. However, it may not need to be treated; that is a benign tumor. On the other hand, some tumors that spread quickly to other parts of the body are called malignant tumors (cancer), which can spread throughout the body and affect your health in different methods [14]. There is no specific way to prevent cancer, but there are factors that reduce the risk of the disease, which are quitting smoking, vaccinations, regular medical examinations, maintaining the ideal weight, exercising regularly, proper nutrition, and early diagnosis [15].

*2.2. Colorectal Cancer*

Colorectal cancer is a serious cancer type and is ranked as one of the top three most deadly and severe cancers in the world after breast cancer and lung cancer [3,4]. Colorectal cancer disease causes many cases of death, and it causes over 4000 people to die annually in the Kingdom of Saudi Arabia [7]. It affects human health by spreading to the lungs, ovaries, liver, and other portions of the digestive system [16].

Colorectal cancer is influenced by numerous factors, including gender, age, medical condition, smoking, alcohol, diet consumption, and genetic disease. Early indications of colorectal cancer include low hemoglobin and changes in bowel habits, such as diarrhea or constipation or a change in stool consistency for more than two weeks, along with bleeding, stomach discomfort, such as gas or pain, abdominal pain, and swelling in the colorectal area [6].

Most people are unaware that the risk of developing colorectal cancer considerably increases between the ages of 40 and 50. In addition, they do not visit the hospital for a checkup for colorectal cancer. Early detection of colorectal cancer provides a possibility for recovery, reduces the risk of dying from colorectal cancer, and enhances the quality of life. Healthcare providers' diagnoses could be inaccurate. For instance, some individuals with abdominal pain are not referred to specialized institutions because their healthcare practitioners identify them as having irritable bowel syndrome. Some healthcare professionals are unaware of the risk factors for colorectal cancer.

There are two main kinds of colorectal exams: FOBT/FIT and colonoscopies. In general, people with modest risks should use FOBT or FIT. If the FOBT/FIT results are negative, it will be done again in a year; if it eventually returns positive, the patient will be recommended for a colonoscopy. Total colonoscopies are recommended for high-risk patients and those who have had a positive FOBT or FIT. The results of a total colonoscopy may be negative, in which case the procedure is repeated every 5 years for low-risk individuals and every year for high-risk individuals, or positive, in which case the patient is directed to therapy [6].

There are four stages of colorectal cancer, including Stages I, II, III, and IV. Several parameters are considered in the approach, including the main tumor's size and location, the amount of its dissemination to lymph nodes and other organs, and the existence of any biomarkers that impact colorectal cancer spread. During certain phases, survival probabilities vary substantially. In the case of colorectal cancer, for instance, more than 94% of patients between the ages of 18 and 65 may survive with effective therapy if diagnosed at Stage I; however, survival rates at later stages are 87%, 74%, and 19%, respectively, [17]. Patients with colorectal cancer have a better chance of survival if their disease is diagnosed and treated immediately after its initial detection. Thus, early detection and treatment are the only ways to prevent cancer-related mortality [18].

### 2.3. Colorectal Cancer Factors

Considering the risk factors for colorectal cancer is essential for early diagnosis. They include age, genetic disease, medical condition, low hemoglobin, and other variables. According to the literature, colorectal cancer is most strongly correlated with low hemoglobin and sudden weight loss [5].

Most of the researchers use age and sex as main factors, such as in papers [19–21] These papers have different goals; some of them aim to classify normal or abnormal states, Ref. [20], and some of them aim to know the survival times, Refs. [19,21]. On the other hand, in paper [5], the size and grade of the tumor are taken as the main factors because the goal was to predict the stage of the tumor and treatment. Colorectal cancer factors can be classified into patients' personal information and clinical presentation [5,22].

- Personal Information: gender, age, medical illness, smoking and alcohol consumption, diet, family diseases, and genetic disease.
- Clinical Presentation: low hemoglobin, abdominal pain, bleeding, and unintended weight loss.

### 2.4. Artificial Intelligence (AI)

AI is the simulation of human intelligence processes by machines, especially computer systems [8]. In computer science, AI is any device or system that is aware of its environment and takes actions to improve its chances of achieving its objectives [23]. Recently, AI has supported the performance of more complex tasks through ML and DL in various fields. ML is one of the subfields of AI. It focuses on analyzing and interpreting patterns and structures in data to enable learning, inference, and decision-making without human interaction. DL is a subfield of ML, which is essentially a neural network with three or more hidden layers. It aims to imitate the activity of the human brain, enabling them to learn from large amounts of data [24]. The uses of ML and DL are unlimited; nevertheless, among the most well-known applications are semantic analysis, prediction, and computer vision [25].

AI has contributed to the advancement of various sectors, including the medical field [26]. Moreover, AI enables the development of accurate and powerful computer-assisted procedures that can successfully diagnose, treat, screen for cancer, and monitor patient prognosis. breast cancer is one of the serious diseases that AI can diagnose. It can also be used to predict colorectal cancer to reduce mortality rates and to predict the tumor's stage in order to select the appropriate treatment technique [27].

#### 2.4.1. Machine Learning (ML)

ML is a subfield of AI, focusing on how computers may learn to carry out tasks or anticipate outcomes without being explicitly programmed [28]. The term was invented by Arthur Samuel in 1959, who described ML as a field of research that provides learning capability to computers without even being explicitly programmed [29]. ML can help in solving many problems, and one of these problems is the classification problem [29]. Classifying a patient as having colorectal cancer disease or not is an example of classification, as shown in Figure 1. In order to conduct this task, a computer program has to learn from a dataset that contains examples of correctly categorized instances of benign and malignant colorectal and develop a model that can generalize beyond these data. A quantitative performance metric, such as accuracy, sensitivity, and specificity, would be used to assess its performance in properly classifying previously unseen cases of colorectal cancer.

ML algorithms can be broadly divided into three groups: supervised, unsupervised, and semi-supervised learning [30]. In supervised learning, the algorithm learns to react more accurately by comparing its output with those that are provided as input after being given a collection of examples or training modules with the correct outputs. Learning from exemplars or learning from examples are other names for supervised learning. Classification and regression tasks are additional categories for supervised learning activi-

ties. Decision tree (DT) and support vector machines (SVM) are examples of supervised algorithms [29,31,32].

Typically, unsupervised learning aims to identify unidentified patterns in the data and use them to infer rules. When the categories of data are unclear, this strategy is useful. The training data in this case are unlabeled. Unsupervised learning refers to the challenge of identifying hidden structures in unlabeled data and is viewed as a statistic-based approach to learning. One of the most popular examples of the unsupervised algorithm is k-means clustering [29,32].

Semi-supervised learning provides a method for combining the strengths of supervised and unsupervised learning. In the first two categories of output, labels are either given for every observation or none are given at all. There may be instances where certain observations are given labels, but the majority of observations are left unlabeled because labeling is expensive and requires specialized human knowledge. Semi-supervised algorithms are the most appropriate for generating models in these circumstances. Classification, regression, and prediction issues can all be solved using semi-supervised learning [30,33].

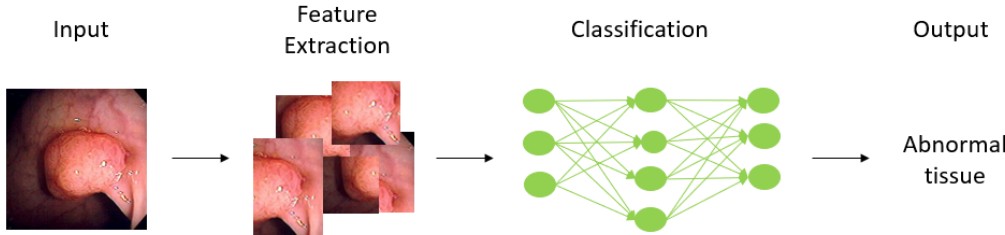

**Figure 1.** An example of colorectal cancer classification using ML.

### 2.4.2. Deep Learning (DL)

Deep learning, a subfield of ML, is the ability of computers to learn from their experiences and to conceptualize the world in terms of a hierarchy of concepts, with each notion defined in reference to simpler concepts [34]. DL reduces the need for human operators to expressly specify every piece of knowledge that the computer needs by learning from experience. By constructing complex ideas out of smaller ones, the hierarchy of concepts enables computers to learn intricate ideas. Although "deep learning" is a relatively recent name, the field has existed since the 1950s [34]. Furthermore, it has been proven that DL has produced excellent outcomes in a variety of fields, including voice recognition, object detection, and medical imaging [3,35]. Figure 2 illustrates how the colorectal area is classified as normal or abnormal by DL.

There are three primary categories of DL architectures that may be categorized by the type of application they are used for supervised deep networks, unsupervised deep networks, and hybrid deep networks. Supervised learning refers to a learning process in which a model is improved by being subjected to labeled data. The majority of medical image analysis applications use convolutional neural networks (CNNs), a supervised learning architecture [36,37].

Unsupervised learning is known as the ability of a computer to identify characteristics from input data without labeled data. Part of unsupervised learning is autoencoders. Both CNN and autoencoders are popular and effective DL models for colorectal cancer. Hybrid deep networks are built by combining various DL architectures to obtain superior results [3].

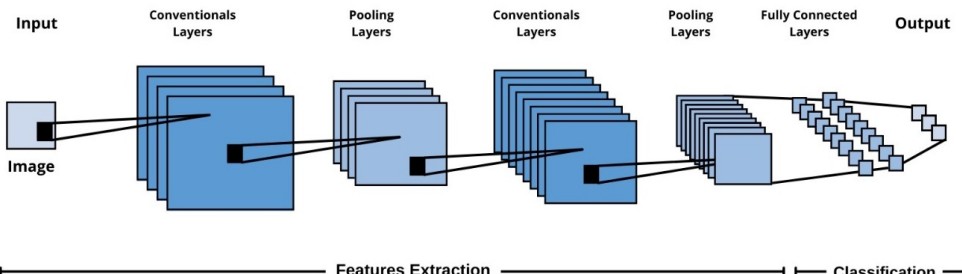

**Figure 2.** Classification of colorectal cancer using DL. Adopted from [10].

## 3. Literature Selection Methodology

This section presents the research articles that were filtered to select the most relevant studies on colorectal cancer prediction using ML and DL algorithms, as summarized in Figure 3. These phases consist of (1) keyword filtering, (2) abstract filtering, and (3) full reading filtering.

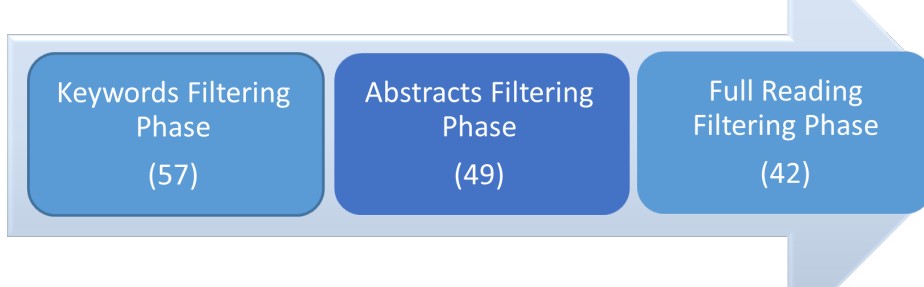

**Figure 3.** Filtering phases of the related research articles.

### 3.1. Keywords Filtering Phase

The related research articles have been collected taking into consideration that the title of the research article contains at least one of these keywords: (1) detect or predict colon or colorectal cancer using ML, (2) detect or predict colon or colorectal cancer using DL, and (3) detect or predict colon or colorectal cancer using AI. This phase resulted in 57 research articles.

### 3.2. Abstract Filtering Phase

Beginning with the collection of 57 research articles, an abstract reading was conducted to select only the most relevant articles to the topic, which is the implementation of AI algorithms for predicting colorectal cancer. Then, 49 research articles were chosen from 57 research articles.

### 3.3. Full Reading Filtering Phase

In consideration of the previous phase's 49 research papers, an all-article reading was conducted to determine the most relevant research articles that explicitly attempt to predict colorectal cancer using AI. Hence, this phase reduces the number of research publications from 49 to 42.

## 4. Taxonomy of Literature Reviews

As indicated in Figure 4, the reviewed study articles are categorized as follows: (1) the aim of the prediction, (2) the method of the prediction, and (3) the dataset used in the prediction.

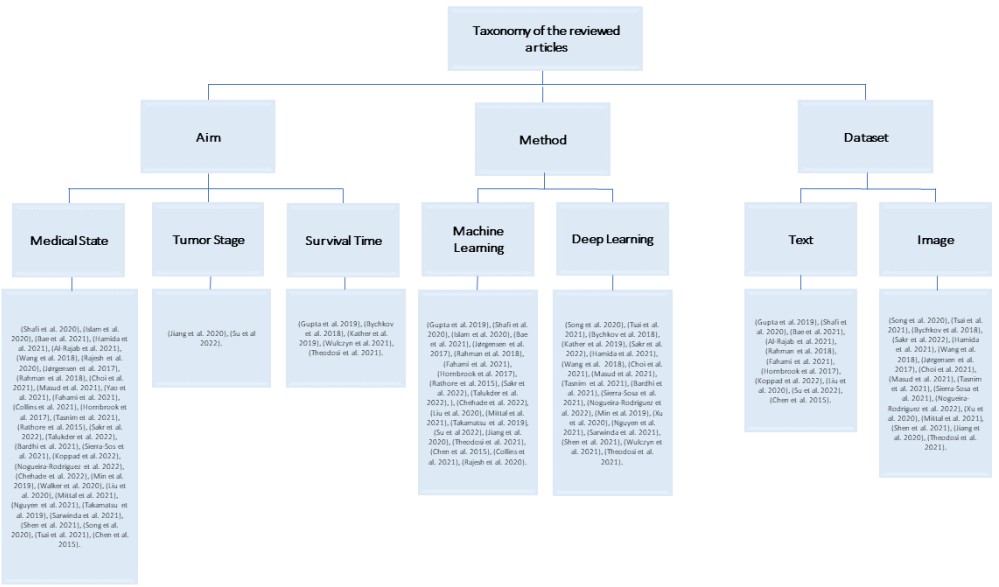

**Figure 4.** Taxonomy of the reviewed articles on predicting colorectal cancer using AI [5,9,10,19–22,38–72].

### 4.1. Based on the Aim of Prediction

Most AI models for the prediction of colorectal cancer are classified in terms of the aim of building the model into (1) predicting the state of patients, whether normal or abnormal, (2) predicting the tumor stage, and (3) predicting survival time. The first aim targets the early detection of colorectal cancer, and the other two aims target patients who have been diagnosed with colorectal cancer.

4.1.1. Aim 1: To Predict Medical State

In [38], the authors used ML algorithms and feature selection techniques to detect colon cancer. The Malondialdehyde (MDA) and maximum degree greedy (MDG) algorithms were used for feature selection. The RF, SVM, logistic regression (LR), AdaBoost, and KNN algorithms have been applied to a public dataset that is made up of 62 cases and 2000 genes. There are 40 abnormal and 22 normal patients among them. The result showed that the RF algorithm with the feature selection method achieved the highest accuracy with 95.161%. The model deals only with genes as features.

In [39], the study aims to classify tissues into normal and abnormal using an ensemble classifier method. The feature selection methods that were used are filtering and wrapping. ML algorithms, which are RF, KSVM, eXtreme Gradient Boosting (XGB), KNN, and ensemble, were applied to 62 patients and 1200 gene expressions at the Bioinformatics Research Group of Pablo de Olavide University. There are 40 abnormal and 22 normal patients among them. The finding of this model was that the ensemble method achieved a higher accuracy of 91.67%.

In [40], the authors looked at the problem of detecting the presence of colorectal cancer. The main algorithm applied was the modified Harmony Search Algorithm (Z-FS-KM-MHS). A total of 2000 genes from the Princeton University Gene Expression Project were used. The results showed that Z-FS-KM-MHS achieved accuracy up to 94.36%. The model used a large number of genes. Unlike other studies, this method can be applied to diseases related to genes, such as breast cancer.

Hamida et al. proposed a model for classifying colon images into normal or non-normal using the convolutional neural networks (CNN) algorithm. The main models applied were the UNET and SEGNET models for 100,000 histopathological images in Germany. SEGNET reached a high-performance accuracy of 99.5%. The authors concluded that DL is better for classifying images than ML for handling large-scale images. Unlike [38–40], images were used to classify colon cancer [41].

In [42], the authors improved the diagnosis of colon cancer. The model applied ML algorithms, which are SVM, naive Bayes (NB), decision trees (DT), and KNN to two public datasets containing 9457 genes and 98 samples. KNN and DT were the best in classification using the first dataset, and NB was the best in classification when using the second dataset.

In [43], the authors consider the problem of not detecting polyps through colonoscopy. The main method applied was DL in 1290 patients and 27,113 colonoscopy images from the Endoscopy Center of Sichuan Provincial People's Hospital. The algorithm achieved a per-image sensitivity of 91.64%. However, the algorithm detects only polyps.

Rajesh et al. proposed a Density-Based Spatial Clustering of Applications with Noise (DB-SCAN) algorithm for colon tumor detection from biopsy samples by categorizing the normal or harmful cells. The algorithm was tested on 100 images collected from Zendo repositories. The results showed that the model achieved 99% accuracy, 85.4% sensitivity, and 87.6% specificity in detecting colon tumors [44].

Jørgensen et al. used cell nuclei to extract information for the detection of cancerous tissue, whether it is benign or cancerous. This algorithm consists of RF, color deconvolution, k-means clustering, local adaptive thresholding, and cell separation within the region of interest (ROI) on 87 colon tissue slides. As a result, the algorithm obtained an area under the curve (AUC), sensitivity, specificity, and accuracy of 0.96, 0.88, 0.92, and 0.91, respectively, [45].

Akizor and Ravi proposed a model for classifying lung and colon cancer using ANNs by applying a feature selection method. In this model, authors have used a public dataset consisting of 2000 genes and 62 instances. The classification accuracy reached 98.4% for the two classes of normal and cancer. In addition, the authors have found that using the feature selection method may increase the classification accuracy of the model. However, the sample size was small [46].

Choi et al. built a computer-aided diagnosis (CAD) system by applying a DL model to predict four categories: normal, low-grade dysplasia, high-grade dysplasia, and adenocarcinoma of the pathologic histology of colorectal adenoma. The model applied CNN's algorithm using 3400 computed tomography (CT) images collected from Korea Anam University Hospital (KUMC) and a CAD to develop a diagnostic system that predicts tissue adenoma of the colon and rectum. Then, the authors compared the results of the system with the results of the experts. The results showed a classification with a specificity of 92.42% and a sensitivity of 77.25%. In addition, it was close to the results of the experts. One of the limitations the authors faced is that there are not enough samples to assess the validity of their model [47].

Yao et al. proposed automated classification and segmentation of colorectal images based on the self-speed transmission network into three classes: normal tissue, polyps, and tumors. A pre-trained network on ImageNet was first applied to improve the result, and then it was applied to 3061 images. For segmentation, the Unet network framework was used, and then the trained Self-paced Transfer VGG (STVGG) network model was used for colorectal classification. The model has obtained high accuracy for both classification and segmentation. This paper is distinguished from the rest of the studies in that it combined two objectives, namely classification and segmentation. It also used self-learning to solve the problem of imbalance, learn the difficult sample, and raise performance [48].

Masud et al. proposed a classification framework using digital image processing (DIP) and modern DL. Features were extracted from 25,000 pathological histological images using DIP. After that, features were collected and categorized using the CNN model. The classification of lung and colon cancer demonstrated an accuracy of 96.33%, a precision of 96.39%, a recall of 96.37%, and an f-Measure of 96.38%. This study mentioned in detail an important point, which is to process the images before implementing the algorithm using two techniques, and this may help to raise the accuracy of the results [49].

Fahami et al. proposed a model to detect the most effective genes for colon cancer patients in their vital status by using ML methods such as neural networks, KNN, and DT. As a result, DT has high accuracy when using the HTSeq-FPKM-UQ public dataset [50].

Collins et al. discussed how to detect colorectal and esophagogastric cancer tissue using an automatic CAD tool via an optical image. The dataset is from a public university hospital in Leipzig and includes 10 patients with Esophagogastric cancer and 12 patients with colon cancer. After studying the receiver operator curve–area under the curve (ROC–AUC) performance in the colon dataset in different models such as radial basis function (RBF)–SVM, MLP, and 3-dimensional convolutional neural network (3D CNN), it has been concluded that the 3D CNN model achieved a more accurate performance to detect colon cancer with an accuracy of 93% [20].

Hornbrook et al. suggested that in the United States (US), a community-based, insured adult population can detect colorectal cancer by using ML and making a diagnosis based on gender, age, and complete blood count data. The dataset used is Kaiser Permanente Northwest Region's Tumor Registry (KPNW), which includes 439 females and 461 males. The result of high-risk detection with colorectal cancer was 99% [51].

Chen et al. developed an innovative deep learning algorithm for shallow neural networks (SNN). They present an SNN model with a set of parameters in the supervised model, and they decompose its computational process into a number of positive parts that work smoothly together to produce a superior outcome. It produces consistently high-quality results, on par with those of deep neural networks. In addition to describing the algorithm, this paper analyzes it.

In [52], the authors improved an approach that takes around a minute to identify colon cancer from the input picture using a CNN algorithm with the max pooling and average pooling layers and MobileNetV2 models. The dataset contains 25,000 images taken from the Kaggle platform. The accuracy results were 97.49% and 95.48%. Moreover, the MobileNetV2 model has the highest accuracy rate of 99.67%.

Rathore et al. developed a model for predicting cancer in colon tissues using a Hybrid Feature Space-Based Colon Classification (HFS-CC) technique and the k-means algorithm for clustering. 68 colon biopsy samples were taken from randomly selected patients at Rawalpindi Medical College (RMC). The result showed that the HFS-CC technique achieves 98.07% accuracy [53].

In [22], the authors proposed a new efficient method for the detection of colon cancer using DL. In order to detect colon cancer, the CNN algorithm with the LC25000 Lung and Colon Histopathological Image dataset has been used. The result showed that the CNN algorithm provides a high accuracy of 99.50%. However, this paper did not employ optimization techniques to select the best features from the extracted deep features.

In [54], the authors developed a hybrid ensemble feature extraction model to efficiently identify lung and colon cancer. The method that has been used is feature extraction and ML algorithms with the LC25000 Lung and Colon Histopathological Image dataset. The accuracy rate for lung cancer detection was 99.05% and for colon cancer it was 100%; for lung and colon cancers, it was 99.30%. Even though the outcome of detecting lung and colon cancer has high accuracy, the model needs to be evaluated using a different dataset to see how it performs.

The authors in [55] used the combination of CNN and autoencoders on the CVC-ColonDB, CVC-ClinicDB, and ETIS-LaribPolypDB datasets. The proposed model achieved 96.7% accuracy, which is better than the models compared to it: the deep CNNs model,

which has an accuracy of 96.4% and the AI-Assisted Polyp Detection model, which has an accuracy of 76.5%, although not very significant.

In [56], the application of DL techniques for the detection and segmentation of colon polyps in colonoscopies has been presented. The authors used region-based convolutional neural networks (R-CNN), path aggregation network (PA-Net), Cascade R-CNN, and Hybrid Task Cascade (HTC) with ClinicDB, ETIS, and Deusto University e-Vida research group datasets. The outcome showed that the best detection rate was acquired when training the model with all the datasets and using PANet architecture. The best segmentation accuracy was acquired when using HTC architecture trained with the merged dataset and tested on the CVC-CLINIC dataset. On the other hand, the model needs a framework for real-time processing of the live feed from the colonoscopy.

In [57], the study set out to identify genes' associations with colorectal cancer using a cyclic redundancy check (CRC) that can potentially be used as diagnostic markers in translational research. Different algorithms were implemented, which are Adaboost, ExtraTrees, LR, NB, RF, and XGB on the GSE44861, GSE20916, and GSE113513 datasets. The findings showed that 34 genes with high accuracy can be used as a diagnostic panel for CRC. Furthermore, RF achieved an accuracy of 98.2% and had the best performance among all classifiers when using GSE44861 as training data and GSE20916 as test data. The results of this research can aid in the identification of risk factors for colorectal cancer; while ensuring the clinical utility of the indicators discovered, it will help doctors make more informed decisions about treating colorectal cancer. However, specific trials are needed to validate the findings of this study.

The authors in [58] tested a previously published polyp detection model with ten public colonoscopy image datasets. The You Only Look Once version 3 (YOLOv3) model has been used with ten datasets, which are: CVC-ClinicDB, CVC-ColonDB, CVC-PolypHD, ETIS-Larib, Kvasir-SEG, CVC- ClinicVideoDB, PICCOLO, KUMC dataset, SUN, and LD-PolypVideo. The paper showed that when evaluating the recently published model on a private test partition, the F1-score was 0.88. When tested on ten public datasets, it decayed by 13.65% on average. The authors pointed out the interest in comparing intradataset performances (i.e., a performance evaluation on a test split of the dataset used for model development, either private or public) versus interdataset performances (i.e., a performance evaluation on a dataset different from the one used for model development). The authors in [59] constructed an automated system that can accurately classify the subtypes of colon and lung cancer. They used the Lung and Colon Histopathological Image (LC25000) datasets with ML, feature engineering, and image processing approaches. According to the results, the XGBoost has an F1 score of 98.8% and an accuracy of 99%.

Min et al. built a computer-aided diagnosis (CAD) system by applying linked color imaging (LCI) images to predict the histological results of polyps, whether they are adenomatous or non-adenomatous. The CAD system was trained and tested in this study with a dataset from the Hospital of the Academy of Military Medical Sciences. There were 389 images and 203 patients in the dataset. Due to the small size of the dataset, the authors used a Gaussian mixture model (GMM) to train the system. The system was therefore accurate to 78.4%, a specificity of 70.1%, a sensitivity of 83.3%, and PPV of 82.6%. The accuracy of the CAD system was comparable to that of expert endoscopists when compared to them [60].

In [61], the authors proposed a model for colorectal cancer detection based on DL. The main algorithm applied is CNNs in 322 images from St. Paul's Hospital. The CNN algorithm achieved an accuracy of 91.64% for normal slides and 94.8% for cancer slides. In training, heavy data augmentation was performed to increase the robustness of the model.

In [62], the authors aimed to identify the fundamental transcript factors (TFs) associated with the clinical outcomes of colon cancer patients by combining the random forest algorithm with the traditional Cox proportional hazard (Cox PH) method. The system used public datasets from the GEO database, which include 925 patients with colon cancer. As a result, the authors were able to construct a predictive model for the prognosis signature of colon cancer and successfully identify five TF signatures.

Mittal et al. developed a new classification method that, by combining MALDI-MSI with supervised ML, can accurately predict lymph node metastasis (LNM) status for patients with primary endometrial cancer (EC) and differentiate between colorectal cancer (CRC) and normal tissue. In this model, the authors classified by using neural networks (NN). The study made use of a dataset from the PRIDE partner repository, which contained 15 TMAs and 302 patients' images related to CRC and EC. The model correctly identified the metastasis stage of approximately 80% of the EC spectra and 98% of the CRC spectra as being derived from normal or tumorous tissue [63].

In [10], the researchers aimed to use DL technology to identify medical images to increase the accuracy of the identification due to the automatic classification of tumor types. The authors used CNN with the NCT-CRC-HE-100K, Kather-texture-2016-image, and CRC-VAL-HE-7K datasets. The study's findings showed an accuracy rate of 99.69% when using NCT-CRC-HE-100K and a rate of 99.32% when using CRC-VAL-HE-7K.

The goal of [9] was to establish a semantic segmentation model for the diagnosis of colorectal adenomas. The dataset was obtained from the Chinese People's Liberation Army (PLA) General Hospital (PLAGH), the Cancer Hospital, the Chinese Academy of Medical Sciences (CH), and the China–Japan Friendship Hospital (CJFH) and used with deep CNN. The results revealed that accuracy had reached 90%.

In [64], this study aims to classify colorectal tissue images, including tumor versus normal tissues and other tissue types, by TMA core images, which is a treasure trove for artificial intelligence applications. Using DL techniques, TMA tissue cores can be classified into five classification flows: the two first classification flows, each NN (VGG16 and CapsNet), and the three other classification flows are based on ensemble methods. The performance metrics used are recall, precision, F1-score, and accuracy. In another hand, the dataset contains 770 patients from three different cohorts, which included 410 Swiss patients, 89 German patients, and 271 Canadian patients, along with 54 TMA slides. The best results achieved in this paper were from a Soft Voting Ensemble comprising one VGG and one CapsNet model, with a prediction accuracy of 0.939, 0.982, and 0.947 for tumor, normal, and "other", respectively.

In [65], the aim was to develop a prediction model of lymph node metastasis (LNM) based on ML. The dataset used in this study is available in the Figshare repository. For the training dataset, which consisted of 277 slide images, a random forest algorithm was used, and 120 slide images were used for the test. The findings demonstrate that ML has a high predictive value of 0.938 AUCs.

Sarwinda et al. proposed a model for the detection of colorectal cancer. The method of the model is DL using residual network (ResNet). There are two classifiers, which are ResNet-18 and Resnet-50. The authors have trained ResNet-18 and ResNet-50 on colon gland images to distinguish colorectal cancer from benign and malignant. The dataset from a Warwick-QU consists of 165 images, including 74 benign tumor images and 91 malignant tumor images. ResNet-50 achieved the highest accuracy of 88% when testing data values of 20% and 25%. When the training data increase, the accuracy of the model increases [66].

Ref. [67] suggested a high-throughput system to precisely identify tumor areas on colorectal cancer histology slides. The methodology that has been used is a CNN model and a Monte Carlo (MC) adaptive sampling method with three datasets of colorectal cancer from The Cancer Genome Atlas (TCGA). The result achieved an accuracy of 98.90%.

### 4.1.2. Aim 2: To Predict the Tumor Stage

In [68], the authors considered the problem of the diagnosis of colon cancer and its staging. This study used weighted gene co-expression network analysis (WGCNA), the least absolute shrinkage and selection operator (LASSO) algorithm, survival analysis, RF, SVM, DT, and differentially expressed genes. The dataset used was the Cancer Genome Atlas (TCGA) dataset. The findings showed that RF reached a 99.81% accuracy in the diagnosis of colon cancer compared to SVM and DT. Besides that, RF had an average accuracy of 85.49% for the diagnosis of colon cancer staging compared to SVM and DT. In contrast, disclosure of the factors that affect the classification of the three ML methods

has not been identified. Moreover, the staging diagnosis accuracy of the model is low and needs to be improved.

In [69], this study proposed using limited biomarkers as predictors of colon cancer in Stage III by combining CNN with a machine classifier using routinely Hematoxylin and Eosin (H&E)-stained tissue slides. The dataset from West China Hospital (WCH) was randomly split into two sets, including 101 for training the classifier and 67 for validation. The results of this model (Gradient Boosting-Colon) provided a Hazard Ratio (HR) for high-risk and low-risk recurrence of 8.976 (95%).

4.1.3. Aim 3: To Predict Survival Time

Pushpanjali et al. proposed a model for determining the stage of the tumor and the survival time of colon cancer patients. ML algorithms, which are RF, SVM, LR, AdaBoost, and KNN, were applied to 4021 patients at Memorial Hospital. The dataset contains tumor size, tumor grade, and tissue. The best performance algorithm was RF which achieved the highest accuracy in predicting the 5-year depth-first search (DFS) of colon cancer patients and predicting the stage of the tumor [5].

In [70], the authors developed a deep learning system (DLS) for predicting disease-specific survival (DSS) in Stage II and Stage III colorectal cancer. DLS consists of a tumor segmentation model and a prediction model. Furthermore, they applied ANN and CNN algorithms in these models. The authors used a dataset of 27,300 slides collected from the Medical University of Graz. The model achieved an AUC of 0.70 (95% CI: 0.66–0.73) in dataset 1 and 0.69 (95% CI: 0.64–0.72) in dataset 2 and identified potential new predictive features.

Kather et al. proposed a model for predicting survival for colorectal cancer patients. CNN's DL algorithm was utilized as the model. It also used 100,000 histological images from two public datasets, "Darmkrebs: Chancen der Verhütung durch Screening" (DACHS) and TCGA to predict survival. There are two major tissue classes: tumors and stroma. As a result, the CNN algorithm achieved an accuracy of 94%. CNN can predict directly from histopathological images [21].

Bychkov et al. proposed a model for predicting 5-year disease-specific survival for colorectal cancer patients based on tumor tissue images. The model applied the DL algorithm, which is CNN, to 420 patients at the Helsinki University Central Hospital. In the result, the long short-term memory (LSTM) classification achieved an AUC accuracy average of 69%. DL algorithms deal with all image sizes effectively and flexibly [19].

In [71], the authors showed that IHC-stained images of the amplified breast cancer 1 (AIB1) protein from CRC patients could operate as a predictive 5-year survival marker. The dataset was from the University Hospital of Patras, Greece, and contained biopsy material from 54 patients with diagnosed CRC. In addition, they used a pre-trained CNN VGG16 to extract DL features, an SVM classifier, and a bootstrap validation method to enhance the accuracy of 5-year survival prediction. The accuracy of the supervised ML model in predicting 5-year survival was 87%. The classification accuracy of the DL method, which used images at all magnifications, was 97%.

*4.2. Based on the Prediction Method*

In this section, the reviewed studies are classified into two categories based on the prediction method. The algorithms used in predicting colorectal cancer are either ML or DL.

- Machine Learning (ML): [5,20,38–40,42,44–46,50,51,53,54,57,59,62,63,65,68,69,71,72]. Table 1 shows the details of prediction models using ML algorithms.
- Deep Learning (DL): [9,10,19,21,22,41,43,47–49,52,55,56,58,60,61,64,66,67,70,71]. Table 2 shows the details of prediction models using DL algorithms.

**Table 1.** Details of prediction models using ML algorithms.

| | ML | |
|---|---|---|
| **Ref.** | **Algorithm** | **Result** |
| [5] | RF, SVM, LR, AdaBoost, and KNN | ACC, precision, and F-measure of RF: 89% and recall: 88% |
| [38] | RF, SVM, LR, AdaBoost, and KNN | ACC, precision, and recall of RF: 95.16% and F-measure: 95.12% |
| [57] | Adaboost, ExtraTrees, LR, NB, RF, and XGB | RF: ACC: 98.2% |
| [39] | RF, KSVM, XGB, KNN, and ensemble | ensemble: ACC: 91.67%, precision: 82%, recall: 100%, and MCC: 0.85 |
| [40] | Z-FS-KM-MHS | ACC: 94.36% |
| [42] | SVM, NB, DT, and KNN | KNN: 94% using dataset1 and NB has ACC: 100% using dataset2 |
| [45] | RF, color deconvolution, k-means clustering, local adaptive thresholding, and cell separation | AUC: 0.96, sensitivity: 0.88, specificity: 0.92, and ACC: 0.91 |
| [50] | Naive Bayes, QDA, SVM Linear Kernel, LD, AdaBoost, LR, KNN, and DT | DT: ACC: 100% |
| [72] | SNN | ACC: 84% |
| [46] | ANN | ACC: 98.4% |
| [44] | DB-SCAN | ACC: 99%, sensitivity: 85.4%, and specificity: 87.6% |
| [20] | SVM, 3DCNN, and MLP | 3DCNN: AUC: 93% using esophagogastric (EG) dataset |
| [53] | SVMclassifier (linear, RBF, Sigmoid) | ACC: HFS-CC 98.07% |
| [51] | ColonFlag | 34.7 (95% CI 28.9–40.4) |
| [68] | WGCNA, LASSO, survival analysis, RF, SVM, and DT, deferentially expressed genes | RF: ACC: 99.81% |
| [54] | RF, SVM, LR, MLP, XGB | ACC: 100% |
| [59] | SVM, RF, XGBoost, LDA, and MLP | XGBoost: ACC: 95.6% |
| [63] | NN | ACC: 98% |
| [65] | RF | AUCs: 0.938 |
| [62] | Combining RF with Cox PH | Successfully identify five TF signatures |
| [71] | SVM | ACC: 87% |
| [69] | CNN | Hazard Ratio (HR): 95% |

According to Tables 1 and 2, researchers have differed in choosing methods to classify their datasets; 50% of the researchers used ML and 50% of the researchers used DL. In ML, 54.54% of the studies used a text dataset, whereas 45.45% of the studies used a set of images. As for DL, it was applied to a dataset containing images. Whether using ML with a text dataset or an image dataset, it can give high accuracy that reaches 100% based on [50] that used a text dataset and [54] that used an image dataset. However, in [71], ML and DL were used for classification based on an image dataset. The ML model achieved an accuracy of 87% and the DL model achieved an accuracy of 97%, which indicates that DL might be more effective than ML when using an image dataset.

**Table 2.** Details of prediction models using DL algorithms.

| | DL | |
|---|---|---|
| **Ref.** | **Algorithm** | **Result** |
| [47] | CNN | Specificity: 92.42% and sensitivity: 77.25% |
| [48] | STVGG | ACC: 96% |
| [49] | CNN | ACC: 96.33%, precision: 96.39%, recall: 96.37%, and F-measure: 96.38% |
| [41] | CNN | ACC: 99.5% |
| [52] | CNN | ACC: 99.67% |
| [22] | CNN | ACC: 99.50% |
| [55] | CNN and autoencoders | ACC: 96.7% |
| [56] | Mask-RCNN, PANet, Cascade R-CNN, and HTC | PANet: ACC: 97.83% |
| [58] | YOLOv3. | F1-score: 74.35% |
| [43] | Developing a DL algorithm | ACC: 91.64% |
| [9] | CNN | AUC: 92% |
| [67] | CNN | ACC: 98.90% |
| [10] | CNN | NCT-CRC-HE-100K dataset: ACC: 99.69%, CRC-VAL-HE-7K dataset: ACC: 99.32% |
| [70] | ANN and CNN | Dataset 1: AUC: 0.70 (95% CI: 0.66–0.73) Dataset 2: AUC: 0.69 (95% CI: 0.64–0.72) |
| [60] | GMM | ACC: 78.4%, specificity: 70.1%, sensitivity: 83.3%, and PPV: 82.6% |
| [19] | VGG-16 and LSTM | AUC 0.69 |
| [61] | CNNs | ACC: 91.64% for normal slides and 94.8% for cancer slides |
| [21] | CNNs | ACC: 94% |
| [66] | ResNet | ACC: 88% |
| [71] | VGG16 pre-trained CNN | ACC: 97% |
| [64] | Ensemble approach with two CNNs (VGG and CapsuleNet) | Overall average SVEVC: recall: 0.922, precision: 0.907, F1-score: 0.910, ACC: 0.956 |

*4.3. Based on the Type of Dataset*

In this section, the reviewed studies are classified into two categories based on the type of dataset. The most important factor for ML architectures is the process of precisely obtaining the right data. ML architectures must have a suitable and sufficient set of training and test data [3]. As well, it is important to note that training data is a subset of the original data used to train ML models whereas testing data is used to check whether the models are accurate. In general, the training dataset is generally larger compared to the testing dataset [73]. The type of dataset used in predicting colorectal cancer is either a text dataset or an image dataset.

- Text Dataset: [5,38–40,42,46,50,51,57,62,68,72]. The authors used text datasets and they are summarized in Table 3.
- Image Dataset: [9,10,19–22,41,43–45,47–49,52–56,58–61,63–67,69–71]. The authors used image datasets and they are summarized in Tables 4–8.

Figure 5 presents the type of dataset that has been used in the literature. A total of 28.57% of the studies used a text dataset and 71.43% of the studies used an image dataset, as shown in Figure 6. We conclude from the reviewed papers that using a text dataset or an image dataset to train and test the model can have high accuracy in predicting colorectal cancer.

**Table 3.** Details of prediction models using text dataset.

| Ref. | No. of Samples/Patients | Dataset Source | Dataset Availability | Algorithm | Result |
|---|---|---|---|---|---|
| [5] | 4021 patients | Memorial Hospital | Collected | RF, SVM, LR, AdaBoost, and KNN | ACC, precision, and F-measure of RF: 89% and recall: 88% |
| [39] | 62 patients | Bioinformatics Research Group of Pablo de Olavide Universit | Collected | RF, KSVM, XGB, KNN, and ensemble | Ensemble: ACC: 91.67%, precision: 82%, recall: 100%, and MCC: 0.85. |
| [40] | 2000 samples | Princeton University Gene Expression Proje | Collected | Z-FS-KM-MHS | ACC: 94.36% |
| [51] | 1000 patients | KPNW | Public | ColonFlag | Specificity: 99% |
| [46] | 2000 samples | Microarray Dataset | Public | ANN | ACC: 98.4% |
| [38] | 62 patients | Department of Molecular Biology | Public | RF, SVM, LR, AdaBoost, and KNN | CC, precision, and recall of RF: 95.16% and F-measure: 95.12% |
| [72] | 2000 to 50,000 samples | [74] | Public | SNN | ACC: 84% |
| [42] | 98 samples | Department of Molecular Biology | Public | RSVM, NB, DT, and KNN | KNN is 94% using dataset1 and NB has ACC: 100% using dataset2 |
| [68] | 521 samples | TCGA | Public | RF, SVM, and DT | RF: ACC: 99.81% |
| [57] | 229 samples | 111 samples from GSE44861, 90 samples from GSE20916, and 28 samples from GSE1135 | Public | Adaboost, ExtraTrees, LR, NB, RF, and XGB | RF: ACC: 98.2% |
| [50] | 40 samples | HTSeq-FPKM-U | Public | Naive Bayes, QDA, SVM Linear Kernel, LDA, AdaBoost, logistic regression, KNN, and DT | DT: ACC: 100% |
| [62] | 925 patients | GEO | Public | Combining RF with Cox PH | The most important factors are: HOXC9, ZNF 556, HEYL, HOXC4, and HOXC6. |

**Table 4.** Details of prediction models using colonoscopy image dataset.

| Ref. | No. of Samples/Patients | Dataset | Image Type | Dataset Availability | Algorithm | Result |
|---|---|---|---|---|---|---|
| [48] | 3061 samples | Colonoscopy image dataset | Colonoscopies images | Collected | STVGG | ACC: 96% |
| [60] | 389 samples and 203 patients | Colonoscopy image dataset | Colonoscopy images | Collected | GMM | ACC: 78.4%, specificity: 70.1%, sensitivity: 83.3%, and PPV: 82.6% |
| [71] | 162 images and 54 patients | Colonoscopy image dataset | Colonoscopy images | Collected | VGG16 pre-trained CNN, SVM | VGG16 pre-trained CNN ACC: 97%, SVM ACC: 87% |

**Table 4.** *Cont.*

| Ref. | No. of Samples/Patients | Dataset | Image Type | Dataset Availability | Algorithm | Result |
|---|---|---|---|---|---|---|
| [56] | 1210 samples | Colonoscopy image dataset | Colonoscopy images | 402 samples collected, 808 samples were public | Mask-RCNN, PANet, Cascade R-CNN, and HTC | PANet: ACC: 97.83% |
| [58] | 149,644 samples | Colonoscopy image dataset | Colonoscopy images | Public | YOLOv3 | F1-score: 74.35% |
| [43] | 290 patients and 27,113 samples | Colonoscopy image dataset | Colonoscopy images | Collected | Developing a DL algorithm | ACC: 91.64% |
| [55] | 1188 samples | Colonoscopy image dataset | Colorectal polyp images | Public | CNN and autoencoders | ACC: 96.7% |

**Table 5.** Details of prediction models using radiology image dataset.

| Ref. | No. of Samples/Patients | Dataset | Image Type | Dataset Availability | Algorithm | Result |
|---|---|---|---|---|---|---|
| [66] | 165 samples | Radiology image dataset | Colon glands images | Public | ResNet | ACC: 88% |
| [47] | 3400 samples | Radiology image dataset | CT images | Collected | CNN | Specificity: 92.42% and sensitivity: 77.25% |

**Table 6.** Details of prediction models using histopathology image dataset—Part 1.

| Ref. | No. of Samples/Patients | Dataset | Image Type | Dataset Availability | Algorithm | Result |
|---|---|---|---|---|---|---|
| [44] | 100 samples | Histopathology image dataset | Biopsy | Collected | DB-SCAN | ACC: 99% |
| [41] | 5181 patients | Histopathology image dataset | Histopathological images | Collected | CNN | ACC: 99.5% |
| [49] | 25,000 samples | Histopathology image dataset | Histopathological images | Collected | CNN | ACC: 96.33% |
| [53] | 174 images | Histopathology image dataset | Biopsy | Collected | SVM classifier (linear, RBF, Sigmoid) | ACC: HFS-CC 98.07% |
| [45] | 87 samples | Histopathology image dataset | Hematoxylin and eosin (H&E)-stained whole-slide images | Public | RF, color deconvolution, k-means clustering, local adaptive thresholding, and cell separation | AUC: 0.96, sensitivity: 0.88, specificity: 0.92, and ACC: 0.91 |
| [9] | 579 samples | Histopathology image dataset | Histological colorectal images | Collected | CNN | ACC: 92% |
| [70] | 27,300 samples | Histopathology image dataset | Slide images | Collected | NN and CNN | Dataset 1: AUC: 0.70 (95% CI: 0.66–0.73) dataset 2: AUC: 0.69 (95% CI: 0.64–0.72) |
| [19] | 420 patients | Histopathology image dataset | TMAs slides | Collected | VGG-16 and LSTM | AUC 0.69 |

**Table 7.** Details of prediction models using histopathology image dataset—Part 2.

| Ref. | No. of Samples/Patients | Dataset | Image Type | Dataset Availability | Algorithm | Result |
|---|---|---|---|---|---|---|
| [61] | 322 samples | Histopathology image dataset | Hematoxylin and eosin (H&E)-stained whole-slide images | Collected | CNNs | ACC: 91.64% for normal slides and 94.8% for cancer slides |
| [64] | 54 TMA slides and 770 patients | Histopathology image dataset | TMA slides | Collected | Ensemble approach with two CNNs (VGG and CapsuleNet) | Overall average SVEVC: recall: 0.922, precision: 0.907, F1-score: 0.910, ACC: 0.956 |
| [52] | 12,500 samples | Histopathology image dataset | Hematoxylin and eosin (H&E)-stained whole-slide images | Public | CNN | ACC: 99.67% |
| [22] | 10,000 samples | Histopathology image dataset | Histopathological images | Public | CNN | ACC: 99.50% |
| [54] | 25,000 samples | Histopathology image dataset | Histopathological images | Public | RF, SVM, LR, MLP, XGB | ACC: 100% |
| [20] | 22 patients | Histopathology image dataset | Hyperspectral images | Public | SVM, 3DCNN, and MLP | 3DCNN: AUC: 93% using esophagogastric (EG) dataset |
| [59] | 25,000 samples | Histopathology image dataset | Histopathological images | Public | SVM, RF, XGBoost, LDA, and MLP | XGBoost: ACC: 95.6% |
| [67] | 1063 samples | Histopathology image dataset | Hematoxylin and eosin (H&E)-stained whole-slide images | Public | CNN | ACC: 98.90% |

**Table 8.** Details of prediction models using histopathology image dataset—Part 3.

| Ref. | No. of Samples/Patients | Dataset | Image Type | Dataset Availability | Algorithm | Result |
|---|---|---|---|---|---|---|
| [10] | 112,180 samples | Histopathology image dataset | Histopathological images | Public | CNN | NCT-CRC-HE-100K dataset: ACC: 99.69%, CRC-VAL-HE-7K dataset: ACC: 99.32% |
| [63] | 15 samples, 302 patients | Histopathology image dataset | TMA slides | Public | NN | ACC: 98% |
| [65] | 370 samples | Histopathology image dataset | Slide images | Public | RF | AUCs: 0.938 |
| [21] | 908 patients | Histopathology image dataset | Histological images | Public | CNNs | ACC: 94% |
| [69] | 114 samples and 168 patients | Histopathology image dataset | Hematoxylin and eosin (H& E)-stained whole-slide images | Public | CNN | a hazard ratio (HR): 95 |

As shown in Figure 7, 71.43% of the image dataset is divided as follows: 70% of the studies used histopathology images. In addition, 23.3% of the reviewed papers used the colonoscopy image. Moreover, 6.6% of the studies used a radiology image. The kind of image may also have an impact on the classification. The majority of the histopathology images produced results that were more than 90% accurate such as [10,22,52,54]. This indicates that histopathology images contain important features that may significantly aid the classification of images.

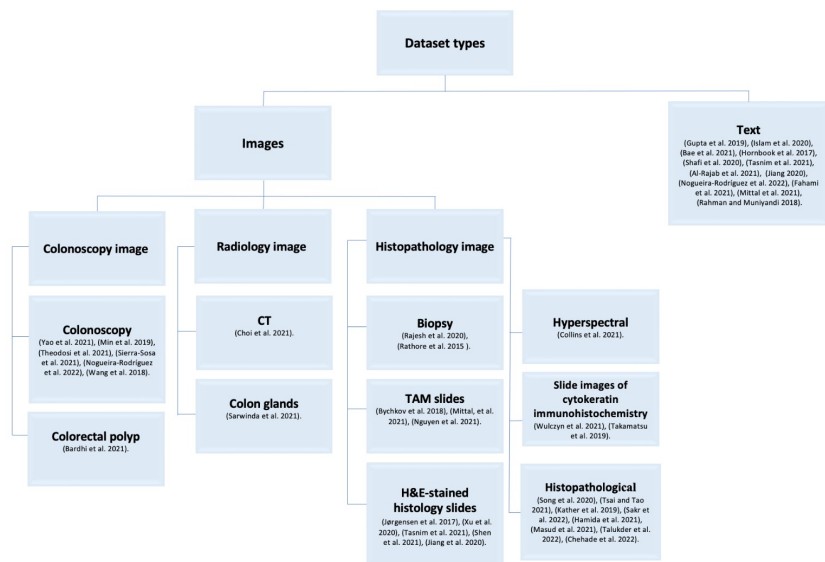

**Figure 5.** Classification of the dataset in reviewed studies articles [5,9,10,19–22,38–72].

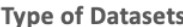

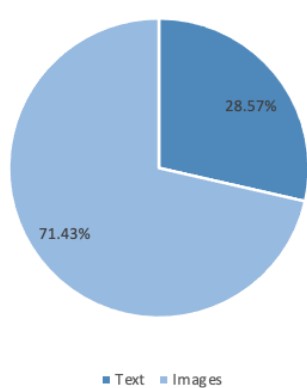

**Figure 6.** Dataset types in terms of text and images.

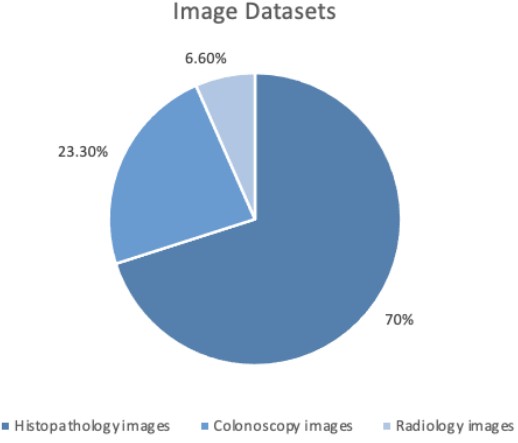

**Figure 7.** Dataset types in terms of images.

Tables 9–11 provide a general summary of the reviewed studies on predicting colorectal cancer using ML and DL algorithms.

**Table 9.** Summary of the reviewed studies on predicting colorectal cancer using ML and DL algorithms—Part 1.

| Ref. | Year | Aim | Classification Problem | Method | No. of Features |
|------|------|-----|------------------------|--------|-----------------|
| [5] | 2019 | Survival time | Binary (patients with DFS greater than 5 years/patients with DFS less than 5 years) | ML | 21 |
| [68] | 2022 | Tumor stage | Binary (tumor tissue/corresponding control tissue) | ML | 19 |
| [38] | 2020 | Medical state | Binary (normal/abnormal) | ML | 2000 genes |
| [39] | 2020 | Medical state | Binary (normal/abnormal) | ML | 1200 genes |
| [40] | 2021 | Medical state | Binary (normal/abnormal) | ML | 20 |
| [50] | 2021 | Medical state | Binary (normal/abnormal) | ML | - |
| [44] | 2021 | Medical state | Binary (normal/ harmful cells ) | ML | - |
| [45] | 2017 | Medical state | Binary (benign/cancer tissue) | ML | 9 |
| [46] | 2018 | Medical state | Binary (normal/abnormal) | ML | 26 |
| [42] | 2021 | Medical state | Binary (normal/tumor) | ML | 57 |
| [51] | 2017 | Medical state | Binary (diagnosed with colorectal cancer, CRC patients with other cancers diagnosed) | ML | 4 |
| [72] | 2015 | Medical state | Binary (normal/abnormal) | ML | 2000 to 50,000 in gene expression |
| [57] | 2022 | Medical state | Binary (normal/CRC) | ML | 10 |
| [20] | 2021 | Medical state | 4-classes: malignant tissue, healthy mucosa of colon, stomach, and esophagus | ML | 7 |
| [53] | 2015 | Medical state | Binary (normal and malignant samples ) | ML | 4 |
| [54] | 2022 | Medical state | 5-classes: benign lung tissue, lung adenocarcinomas, lung squamous cell carcinoma, benign colon tissue, colon adenocarcinomas | ML | - |
| [48] | 2021 | Medical state | 3-classes: normal tissue, polyp, tumor | DL | - |

**Table 10.** Summary of the reviewed studies on predicting colorectal cancer using ML and DL algorithms—Part 2.

| Ref. | Year | Aim | Classification Problem | Method | No. of Features |
|------|------|-----|------------------------|--------|-----------------|
| [47] | 2021 | Medical state | 4-classes: normal, A-LGD, A-HGD, CA | DL | - |
| [49] | 2021 | Medical state | 5-classes: colon adenocarcinoma, colon benign tissue, lung adenocarcinoma, lung benign tissue, lung squamous cell carcinoma | DL | - |
| [22] | 2022 | Medical state | Binary (benign colon/colon adenocarcinomas) | DL | - |
| [55] | 2021 | Medical state | Binary (normal/abnormal) | DL | - |
| [56] | 2021 | Medical state | Binary (normal/abnormal) | DL | - |
| [58] | 2022 | Medical state | Binary (normal/abnormal) | DL | - |
| [52] | 2021 | Medical state | Colon adenocarcinoma (cancer) and colon benign tissue (not cancerous) | DL | - |
| [41] | 2021 | Medical state | Binary (normal/abnormal) | DL | 8 |
| [43] | 2018 | Medical state | Binary (normal/abnormal) | DL | - |
| [9] | 2020 | Medical state | Binary (adenoma/non-neoplasm) | DL | 5 |
| [59] | 2022 | Medical state | 5-classes: colon adenocarcinoma, benign colonic tissue, lung adenocarcinoma, benign lung tissue, and lung squamous cell carcinoma | ML | 37 |
| [67] | 2021 | Medical state | 3-classes: loose non-tumor tissue, dense non-tumor tissue, and gastrointestinal cancer tissues | DL | - |

**Table 11.** Summary of the reviewed studies on predicting colorectal cancer using ML and DL algorithms—Part 3.

| Ref. | Year | Aim | Classification Problem | Method | No. of Features |
|---|---|---|---|---|---|
| [10] | 2021 | Medical state | NCT-CRC-HE-100K and CRC-VAL-HE-7K have 9 classes: ADI, BACK, DEB, LYM, MUC, MUS, NORM, STR, and TUM. Kather-texture-2016-image has 8 classes: TUMOR, STROMA, COMPLEX, LYMPHO, DEBRIS, MUCOSA, ADIPOSE, and EMPTY | DL | 8 |
| [63] | 2021 | Medical state | Binary (normal/tumor tissue) | ML | 1232 |
| [65] | 2017 | Medical state | Binary ( adenomatous/non-adenomatous) | ML | 16 |
| [70] | 2021 | Survival time | Binary (tumor/not tumor) | DL | 200 |
| [60] | 2019 | Medical state | Binary (adenomatous/non-adenomatous) | DL | - |
| [19] | 2018 | Survival time | Binary (low-/high-risk group) | DL | 6 |
| [61] | 2020 | Medical state | Binary (normal/cancer) | DL | - |
| [21] | 2019 | Survival time | Two tissue classes (tumor/stroma) | DL | 3 |
| [62] | 2020 | Medical state | 5-classes (HOXC9, ZNF556, HEYL, HOXC4, and HOXC6 ) | ML | - |
| [71] | 2021 | Survival time | Binary (5-year survivors/non-survivors) | DL and ML | 14 |
| [64] | 2021 | Medical state | 3-classes: (tumor/normal/other tissue) | DL | - |
| [66] | 2021 | Medical state | Binary (benign/malignant) | DL | - |
| [69] | 2020 | Tumor stage | 4 classes: colon cancer into high- and low-risk, and poor and good prognosis groups | ML | 10 |

## 5. Performance Metrics

Performance metrics are an essential part of every ML and DL process. It is used to determine whether or not the model's results are accurate. Several metrics can be used to monitor and evaluate the model's performance. In this part, the metrics employed in the current research publications are discussed.

### 5.1. Accuracy

The basic performance metric is the accuracy that examines the effectiveness of the classifier. In most research articles, the researchers evaluated their model based on accuracy. Accuracy is calculated using Equation (1):

$$Accuracy = \frac{TP + TN}{TP + FP + TN + FN} * 100 \tag{1}$$

True positive (TP) represents the number of correctly classified positive cases. Similarly, true negative (TN) represents the number of correctly identified negative cases. False positive (FP) is the number of actual negative cases that were classified as positive, whereas false negative (FN) is the number of actual positive cases that were classified as negative.

In [44,46,72], the scientists assessed their model in view of accuracy.

### 5.2. Sensitivity

Sensitivity is also known as true positive rate (TPR) or recall. It is the ability of the ML model to accurately identify positive samples. Sensitivity is calculated using Equation (2):

$$Sensitivity = \frac{TP}{TP + FN} * 100 \tag{2}$$

In [39,47,60], the researchers evaluated their model based on a variety of performance measures, including sensitivity.

### 5.3. Precision

Precision is the ratio of the number of correct positive predictions to the total number of positive predictions. Precision is calculated using Equation (3):

$$Precision = \frac{TP}{TP + FP} * 100 \tag{3}$$

In [38,39,49], the researchers used a variety of performance metrics, including precision, to evaluate their model.

### 5.4. F1-Score

The F1-score is a metric to measure a test's accuracy in classification models. In [38,49,58], the researchers assessed their model based on this measure. It is calculated from the precision and recall [3] using Equation (4):

$$F1\text{-}score = 2 * \frac{Precision * Recall}{Precision + Recall} \tag{4}$$

### 5.5. Matthews Correlation Coefficient (MCC)

The Matthews correlation coefficient (MCC) is a statistical measure that is able to accurately reflect the deficiency of any prediction in any dataset. In addition, it is a statistical rate that is more dependable and only gives a high score if the prediction performed well in all four categories of the confusion matrix (TP, FN, TN, and FP) [74]. Models are evaluated based on MCC, such as in [39].

MCC is calculated using Equation (5):

$$MCC = \frac{TP * TN - FP * FN}{(TP + FP) * (TP + FN) * (TN + FP) * (TN + FN)} \tag{5}$$

### 5.6. Area under the ROC Curve (AUC)

The performance of a classification model can be evaluated and contrasted using the area under the curve (AUC). At various probability cutoffs, it is a plot of the proportion of true positives versus false positives. It provides a straightforward means of summarizing a model's overall performance. A worse model is typically one with a significantly lower AUC value the absolute difference between the predicted and actual values. In [20,45,65,70], and [19], the researchers evaluated their models based on the AUC.

## 6. Discussion

In this review paper, 42 research articles were reviewed and summarized in Tables 9–11. The goals of previous research varied, but 83.3% of it aimed to predict the medical state [9,10,20,22,38–63,65–67] and the remaining 15.9% were aimed at predicting the tumor stage [68,69] and survival time [5,19,21,70,71]. On the technical side, 50% of the researchers were using ML algorithms, and 50% of them were using DL algorithms. Regarding the colorectal cancer dataset, 57.2% of researchers used public datasets, which are available online, whereas 42.8% of researchers used their collected datasets. The reviewed studies are divided into three categories, as shown in Figure 8.

It has been found that predicting colorectal cancer using AI algorithms has been conducted in other countries such as Germany [41], America, Korea, and China [48]. Furthermore, these prediction models do not provide a framework that allows the end user to use them. Most models used ML to predict colorectal cancer and the type of dataset is numeric. In addition, it does not focus on predicting colorectal cancer in the early stages, although that point is important to treat colorectal cancer and reduce the risk of colorectal cancer spreading to the rest of the body.

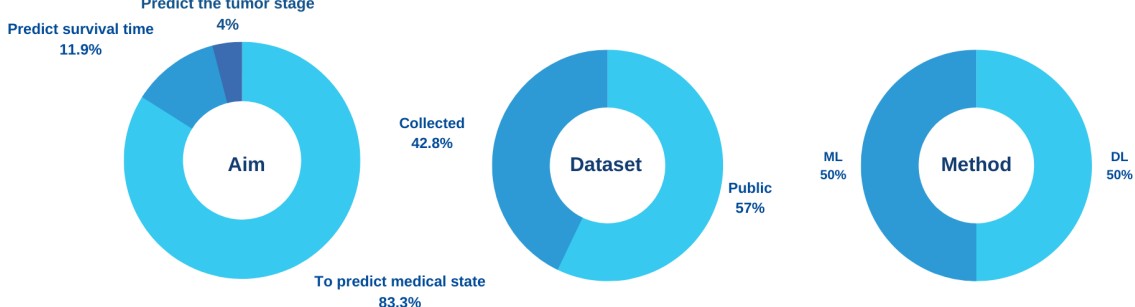

**Figure 8.** Summary of research articles.

## *6.1. Challenges*

There are several challenges in predicting colorectal cancer using AI, which are divided into medical and technical challenges: medical challenges such as lack of awareness and knowledge about AI; technical challenges such as patients' privacy and the reliability of the models.

### 6.1.1. Lack of Awareness

Colorectal cancer is preventable cancer, but it is a type of disease that can be fatal because perhaps people do not want to undergo early detection tests for this type of cancer. Two methods are used to support the process of diagnosing colorectal cancer, including FOBT/FIT and colonoscopy [6]; however, these two methods are tedious and time-consuming. The elderly fear colonoscopies and this inhibits early diagnosis of colorectal cancer. Furthermore, it limits the collection of adequate data for the elderly in the early stages of colorectal cancer disease. AI has increased its performance in diagnosing this disease using ML and DL algorithms.

### 6.1.2. Patients' Privacy

There is sensitivity when collecting patients' data from the hospital because it is one of the patients' rights to have confidence in respecting the privacy and confidentiality of health and social information related to them. Therefore, when obtaining the necessary data to build models, patients' data must be accurately collected. This is considered a major challenge that must be taken into account [75,76].

### 6.1.3. Reliability of the Models

There is no system always capable of delivering perfect results, not even humans. AI-based diagnosis systems may be prone to errors and biases. The results of these models cannot be blindly trusted; they may cause harm to patients if there is misinformation, so efficient algorithms will need to be used [3]. The development of an automated colorectal cancer diagnosis model requires a large dataset to ensure its reliability. The availability of large amounts of data does not obviate the requirement to have data for early colorectal cancer diagnosis. The difficulty in obtaining a large dataset is due to the difficulty in finding participants with early-stage colorectal cancer in older adults (aged 45–70) and the time it takes to collect the necessary data.

### 6.1.4. The Lack of Knowledge about AI

To create AI models designed to predict outcomes, health professionals must have a comprehensive understanding of how algorithms are created, how data sources are evaluated, and how models are built. Collaboration between AI experts and health professionals is needed to effectively implement and use prediction models based on AI.

### 6.2. Opportunities

There are several opportunities for predicting colorectal cancer using ML and DL, which are divided into medical and technical opportunities. Medical opportunities include early detection of the illness. Technical opportunities include improving system security, the development of novel algorithms, real-world system development, and the use of multiple datasets from various sources.

#### 6.2.1. Early Detection of the Illness

Predicting colorectal cancer using ML and a dataset that uses text data can help in the early diagnosis of colorectal cancer because the elderly will not be afraid to do the test using ML and a text dataset because they will not need to refer to a colonoscopy test.

#### 6.2.2. More Efficient System Security

It is essential to put security measures and procedures in place, especially for the systems that will be employed in hospitals or the healthcare sector, and that will support maintaining the privacy and dependability of medical data collection systems. However, because this information is potentially sensitive, it should be kept private by encrypting it and employing authentication procedures to prevent unauthorized access.

#### 6.2.3. Development of Novel Algorithms

There is a lack of algorithms that deal with the different types of data that a dataset contains, such as text and images. For further study, scientists can work on the development of new algorithms that perform well on datasets that contain different types of data [3].

#### 6.2.4. Real-World System Development

Based on the papers that have been reviewed in this paper, there is a lack of developing a real-world system that can be a standard practice in hospitals and the healthcare sector. More research in this field is encouraged to achieve the goal of useful and trustworthy automated systems that can predict colorectal cancer illnesses.

#### 6.2.5. Using Multiple Datasets from Different Sources

The type of dataset is crucial to knowing whether to use DL or ML. Many datasets differ in type; some are text and others are images, but there is a lack of datasets that deal with different types of data. This makes it an attractive field for further study, where researchers can use different resources to gather their data into one dataset and use it for their model [62].

### 7. Conclusions

In this work, we have reviewed 42 recent studies on the application of ML and DL for colorectal cancer detection and diagnosis. To make the review more comprehensible, we collected all the works and classified them into three major categories: the aim of the prediction, the method of the prediction, and the dataset used in the prediction. In each category, we offered summaries of the investigations from several perspectives. For a more in-depth comparison, we arranged the works in tables. We found that most of the studies that have been proposed in recent years focused on developing predictive models using ML or DL approaches aimed at predicting a normal or abnormal state either using a public or collected dataset. Finally, we highlighted technical and medical elements to explore problems and potential in the field of ML and DL applications in colorectal cancer prediction. In conclusion, AI has a major effect on human life in healthcare by improving the prediction of malignancies like colorectal cancer through the use of ML and DL algorithms.

**Author Contributions:** All authors contributed equally to this work. All authors have read and agreed to the published version of the manuscript.

**Funding:** This research received no external funding.

**Conflicts of Interest:** The authors declare no conflict of interest.

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
