# Peer review of "Predicting Colorectal Cancer Using Machine and Deep Learning Algorithms: Challenges and Opportunities"

_2504-2289, doi:10.3390/bdcc7020074_

Round 1

Reviewer 1 Report

Dear Authors,

Paper summary: This review paper summarizes the outcomes of published papers on “predicting colorectal cancer using machine and deep learning”. This is a very important topic but unfortunately, the manuscript is not well organized and lacks the following points.

General Comments:

(1) Organization: Good

- Manuscript has several issues with the scientific writing style and English errors.

(2) Significance: Very High

- Approximately, 5 million people are suffering from colon cancer in 2020. Out of 5, 2 million are new cases. More than 0.93 million people died due to colon cancer. Hence this is an important research topic.

(3) Innovation: Poor

- No novelty as it is a review paper.

(4) Clarity of writing: Good

- Manuscript lacks to explain technically. Please see specific comments below.

Mandatory Changes:

We need several improvements before publishing this manuscript.

Section 4.2: Based on the prediction method: Need to explain the outcome of this section along with the outcome of Tables 1 and 2. If the result has ML and DL mode accuracy and sensitivity information from a different study, what does it means? For example: You need to find out the common study with similar image data sets (CT image of Colon tumor) and find out which ML/DL models perform. You have to categorize all input data and find out meaningful research outcomes.

Section 4.3: You have to categorize different datasets based on MRI, CT, Histopathological images, Biopsy, etc, and find out meaningful outcomes that which modality gives the better result to detect colon cancer.

Tables 3 and 4: Capitalization issue. What is a slide image? Please check thoroughly.

Table5, 6, & 7: Must be explained. Tabulation without explanation in the manuscript is not enough.

Line 27: Why “It” is written as “IT”

Line 56, 58: Throughout the manuscript, it is inconsistent capitalization. In some places, it is colorectal and in other places Colorectal

Line 72: It is not section 1but it is section 2

Line 76: In the layout paragraph, the author forgot to write about section 5.

Line 486: This paragraph has several issues of capitalization, period, and unclear sentences (see highlighted). Authors must proofread the entire manuscript.

This study used Wthe eighted Gene Co-expression Network Analysis

(WGCNA), Least Absolute Shrinkage and Selection Operator algorithm (LASSO), Survival

Analysis, RF, SVM, DT, and differentially expressed genes and the dataset used was The

Cancer Genome Atlas (TCGA) dataset.

model.67 cancers from WCH were used to validate

the predictive effectiveness of the model Moreover, 47 cancers were analyzed from The

Cancer Genome Atlas Colon Adenocarcinoma database from 168 patients.

Author Response

Thank you for your comments. your comments help us in improving the paper and the way that we see the paper. Please find attached point-by-point response letter. 
Regards,

Reviewer 2 Report

This review of colorectal cancer is well researched and provides an extensive overview of the field. I have some minor comments:

1. There should be more references in the "Artificial Intelligence (AI)" subsection of the background. In general there can be additional references in other sections of the ML and DL backgrounds as well.

2. I don't know what Fig. 8 conveys. It seems like it should be more detailed or a more informative graphic, otherwise it can and should be deleted.

As a minor point "Patients Privacy" should read "Patients' Privacy," e.g. at line 649 and Fig. 8.

Author Response

Thank you for your comments. Please find attached the point-by-point response letter. 
Regards,

Round 2

Reviewer 1 Report

Dear Authors,

Thank you for addressing the comments.

You have some minor errors. Please correct it.

-          Figure 3 is before Figure 2. Please correct their position in the manuscript.

-          Table 6, 7, and 8 has the same title “Details of prediction models using histopathology image dataset”. If you divided a long table into 2 parts then add some titles like “part-1, part-2, part-3, etc” or equivalent.

-          Table 9, 10, and 11 has the same title “Summary of the reviewed studies on predicting Colorectal cancer using ML and DL algorithms”. If you divided a long table into 2 parts then add some titles like “part-1, part-2, part-3, etc” or equivalent.

-          Please proofread once again and avoid any minor mistakes like the above.

Good Luck!!!

Author Response

Thank you for your effort and time in reviewing our paper. All comments have been done. 
